# Cruise Industry Trends and Cruise Ships' Navigational Practices in the Central and South Part of the Adriatic East Coast Affecting Navigational Safety and Sustainable Development

**Josip Dorigatti \***, **Tina Perić** 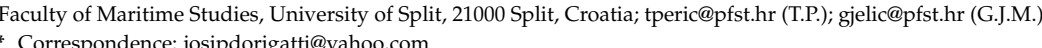 **and Gorana Jelić Mrčelić**

Faculty of Maritime Studies, University of Split, 21000 Split, Croatia; tperic@pfst.hr (T.P.); gjelic@pfst.hr (G.J.M.)
\* Correspondence: josipdorigatti@yahoo.com

**Abstract:** The analysis of cruising trends in the Mediterranean regions shows that the Adriatic is the fastest growing cruise region in terms of the number of passenger movements and cruise ships' port calls among all regions, particularly the central and south part of the east Adriatic coast. The aim of the paper is to analyze leading cruise destination trends in the central and south part of the Adriatic east coast, as well as to identify newly established cruise ships routes, define high-risk navigational and environmental areas and determine cruise traffic density in the vicinity of marine protected areas. The analyses of leading cruise destinations trends are based on four-year (from 2015 to 2019) cruise passenger movement and cruise calls data, whereas analyses of cruise traffic movement are based on one-year cruise ships traffic monitoring (from August 2014 to July 2015). The results of the cruise ship traffic analysis show that cruise ships frequently pass through areas of high navigational and environmental risks that are geographically restricted, navigationally challenging and environmentally sensitive. These routes have become standard navigational practice in newly discovered cruising regions. The obtained results offer a general overview of high-risk cruise ships' navigational practices in coastal navigation that can be associated with any coastal region in the world.

**Keywords:** cruise ships; navigation; safety; sustainability; marine environment; the Adriatic Sea

## 1. Introduction

The popularity of cruise trips has risen since the 1970s [1], and cruise line companies have increased the number of ships as well as the capacities of ships and berths on the market in order to offer broad variety of onboard activities [2]. According to Kovačić and Silveira (2020), the Mediterranean Sea is the most popular cruise destination in Europe (16.7% in 2018) and the second one in the world after the Caribbean (35%) [1]. Italy was the most visited country in the Mediterranean, with 2.4 million cruising passengers, followed by Croatia, with 1.3 million passengers, in 2018 [1]. The analysis of cruising trends in the Mediterranean regions shows that the Adriatic region is the fastest-growing cruise region in the number of passenger movements and cruise ships' port calls [3].

The central and south part of the Adriatic east coast is divided by Croatia, Montenegro and Albania and represents the most valuable natural resource of these countries. Due to its unique beauty and attractiveness, the region is a very popular tourist destination. Natural beauty and cultural–historical diversity are the key factors that attract tourists and represent the main advantage towards competitors [3]. Croatian and Montenegrin economies depend on tourism, as it is their key source of GDP growth. Tourism accounts for some 20% of Croatian GDP [4] and almost 25% of Montenegrin [5].

Globally, the cruising industry experiences higher passenger expectations and higher demands for new bigger cruise ships than ever before. The expansion of the cruising industry is not only related to cruising market expansion but also to the development of new cruising destinations [6]. Due to the strong expansion of cruise traffic and cruise

passenger movement in the central and south part of the Adriatic east coast region, it is necessary to analyze established cruise destinations and new ones as well. The development of new cruise destinations and the increment in the number and the capacity of cruise ships puts cruise ships' navigational decisions in newly established cruising regions with high natural and cultural values under question.

Cruising tourism is not only based on natural beauty and cultural–historical diversity but also on ships' experience and navigation [7]. The longitudinal Adriatic corridor in the northwest–southeast direction and the southeast–northwest direction has greater importance than transversal ones [8–10]. According to the International Maritime Organization (IMO) report Routing of ships, ships reporting and related matters—Establishment of new recommended Traffic Separation Schemes and other routing measures in the Adriatic Sea (IMO, 2015), the north Adriatic region has two separation schemes that efficiently direct maritime traffic. The central and south Adriatic have only one separation scheme (the Central Adriatic Separation Scheme), and it is less regulated by navigational aids than the north Adriatic, therefore offering more options to navigation. The IMO report also elaborated upon the environmental consideration of the traffic in the east Adriatic coast with emphasis on the islands of Vis, Jabuka, Svetac, Biševo, Sv. Andrija, Palagruža and Mljet [11]. The main sailing routes in the Adriatic region were analyzed by Lušić and Kos (2006) [9] and by Zec et al. (2016) [12], and the most attention was given to general longitudinal maritime traffic from the Otranto strait to northern Adriatic ports. The authors stated that maritime traffic flow in the central Adriatic was directed mostly through the Central Adriatic Separation Scheme and that maritime accidents were rare, which indicated good maritime traffic coordination. Lušić et al. (2017) [8] analyzed sailing routes and the structure of maritime traffic in the central part of the Adriatic, with focus on maritime traffic inside the Central Adriatic Separation Scheme. The authors concluded that the depth and width is sufficient on a major part of the longitudinal sailing route and that there were no significant navigational risks except the risk of collision and the risk of grounding near the island of Palagruža.

The aim of the paper is to analyze trends of leading cruise destinations in the central and south part of the Adriatic east coast and to identify newly established cruise ship routes as well as their influence on navigational and environmental safety. One of the main goals of the paper is to determine cruise ships' traffic density in the vicinity of marine protected areas and to identify areas of high navigational and environmental risks. The analyses of leading cruise destinations trends are based on four-year (from 2015 to 2019) cruise passenger movement and cruise calls data, whereas analyses of the longitudinal and transversal traffic flow of cruise ships are based on one-year (from August 2014 to July 2015) cruise ships' traffic monitoring in the central and south part of the Adriatic east coast. The data on cruise passenger movement and cruise calls were obtained and analyzed from a MedCruise report (2019) [10]. The data of cruise passenger movement, cruise calls and cruise ships' movement for years 2020 and 2021 are not taken into consideration due to the COVID pandemic.

### 1.1. Global Cruise Trends

A major part of cruising industry is seasonal, which means that most cruise ships follow the season and shift from one region to another in order to achieve optimal passenger occupancy and offer attractive itineraries to passengers. In order to satisfy passenger demand for exciting itineraries and interesting destinations, cruise ships spend the majority of their time in coastal navigation in areas of high natural, environmental and preservation values. They select the most attractive world destinations and navigate along the most attractive coastlines, many of which are environmentally preserved regions with rich biodiversity and high national importance. The usual cruise industry concept is that cruise ships' operation is regional, and itineraries are repetitive, usually in a circular pattern, with limited durations, offering different port experiences every day.

The question of cruise routes' sustainability in costal navigation becomes important due to the fact that cruise ships spend the majority of their time in coastal navigation. In addition to that, globally, the cruising industry records high passenger demand, high demand for new cruise ships and a trend in which cruise ships are becoming bigger. The expansion of the cruising industry is related to cruising market expansion and the development of new cruising destinations.

Figure 1 shows the relation between cruise ships' number and cruise ships' berths. Cruise ship berth in this context represents a bed of any type on a cruise ship [13]. For the period from 2000 to 2020, the relation between the passenger movement trend (+5.5%) and the cruise ships' berths trend (+4.7%) indicates a higher cruise ships' occupancy rate. On the other hand, the relation between cruise ships' number trend (+4.1%) and cruise ships' passenger movement trend (+5.5%) shows that cruise ships are becoming bigger. Predictions from the 2020 to 2027 period estimate that the number of cruise ships will grow at an average rate of 2.48% and the cruise ships' berths will grow at an average rate of 3.67% [10,13]. Growing trends will slightly stabilize, but they still indicate the delivery of larger cruise ships on the market in order to accommodate high passenger demand.

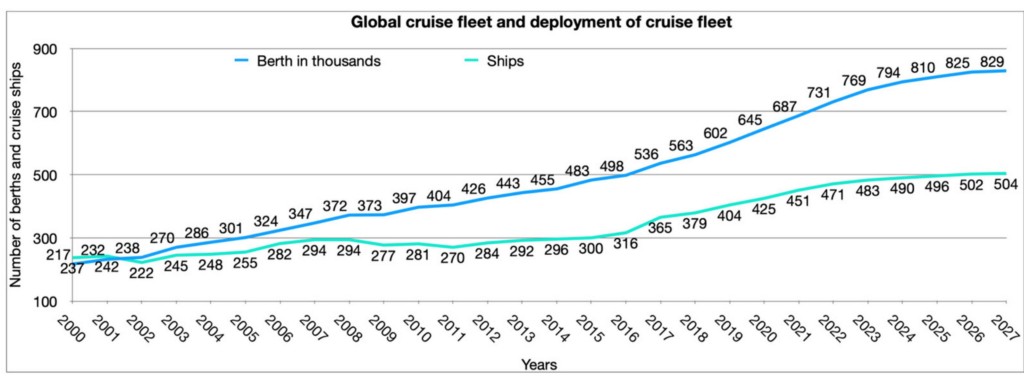

**Figure 1.** Global cruise fleet and deployment of cruise fleet. Adopted from Cruise Industry News 2020–2021.

These factors predict further cruise industry development and cruise traffic expansion. In order to satisfy the demand of the cruise market, it is predicted that popular cruising destinations which are close to reaching their full capacity will maintain the present status or even adjust it, while less prominent cruising regions will continue to grow. It is expected that future cruise industry expansion will focus on new regions that have never been cruising destinations before.

### 1.2. Cruise Trends in the Central and South Part of the Adriatic East Coast

The Mediterranean region is divided into four sub regions (Figure 2), among which, the Adriatic is the second most visited region after the western Mediterranean region [14]. The Adriatic Sea is located between the West Mediterranean and East Mediterranean zone, in proximity to the central and north European market with high cruising demand and developed infrastructure. Due to the geographical location, infrastructural investment and the further expansion of established destinations, the Adriatic region has promising perspectives in future [1].

Rich cultural heritage, historical value, diversity, natural beauty and prominent traditional touristic locations have established well-known cruising destinations such as Venice (1.56 million passengers) and Dubrovnik (0.732 million passengers) which have been market leaders for a long time [1]. The popularity of Venice and Dubrovnik makes cruise shipping in the Adriatic heavily dependent on these two prominent destinations, in contrast to Rodrigue and Notteboom's (2013) statement that destinations do not sell the cruise industry but itineraries [15]. As a result, these two biggest cruise ports in the Adriatic struggle with city congestion.

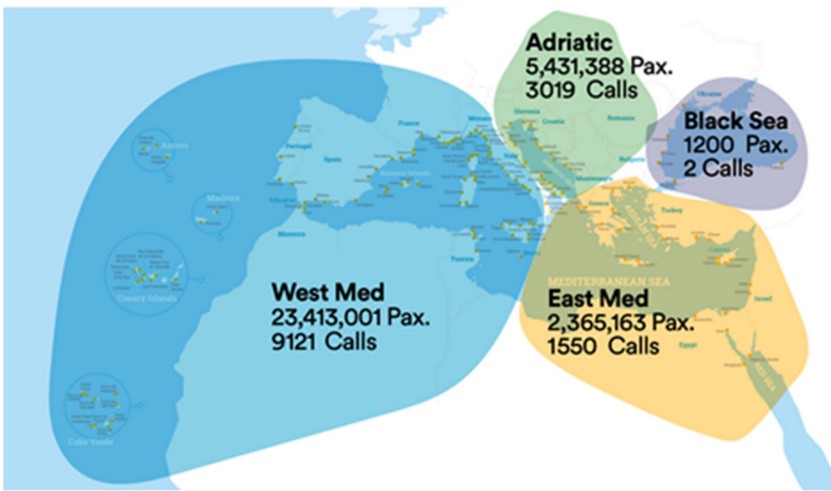

**Figure 2.** Mediterranean cruise market sub regions [11].

The central and south part of the Adriatic east coast extends from the port of Zadar in North Dalmatia to the Otranto strait at the south of the Albanian coast (Figure 3). Croatian and Montenegrin destinations are the most important cruising destinations in this area, while Albanian cruising destinations have not been recognized by the market yet.

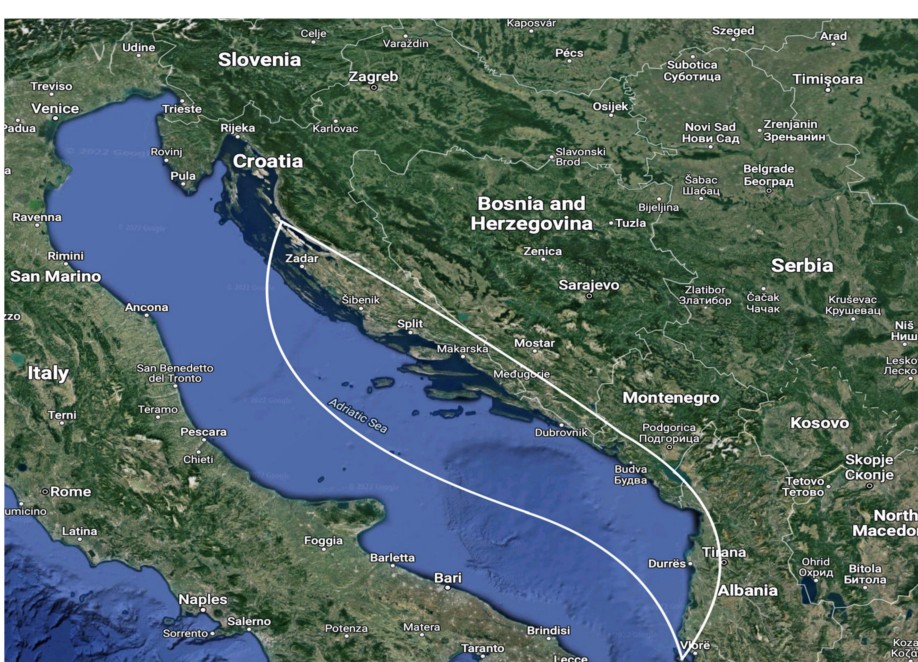

**Figure 3.** The central and south part of the east Adriatic coast. Adopted from Google maps.

The area is recognized worldwide for its natural beauty, cultural heritage and authentic, well-preserved ambient. Many marine protected areas have been established here, such as the Kornati National Park, Mljet National Park, Telašćica Nature Park, the Lastovo Islands Nature Park in Croatia, as well as the Tivat Saline Special Nature Reserve in Montenegro. The United Nations Educational, Scientific and Cultural Organization (UNESCO) has recognized the rich history of this area. The UNESCO World Heritage Sites located in the central and south part of the east Adriatic coast are: the Cathedral of St James in Šibenik, the Historic City of Trogir, the Historical Complex of Split with the Palace of Diocletian, the Stari Grad Plain on the island of Hvar and the Old City of Dubrovnik in Croatia, as well as Natural and Cultural–Historical Region of Kotor in Montenegro and Venetian Works

of Defence between the 16th and 17th Centuries: Stato da Terra—Western Stato da Mar in Croatia and Montenegro [11].

The number of cruise ships' ports in the Adriatic region is constantly rising; currently, around 30 cruise ports are in use by cruise lines [13]. The Adriatic Sea is the fastest growing region in the Mediterranean, with 18.70% growth in cruise calls (from 2492 cruise calls to 3019 cruise calls) and 20.90% growth in total passenger movements (from 3084 cruise passenger movements to 3748) in the period from 2015 to 2019 [10].

Figure 4 shows total cruise call trends from 2015 to 2019 in five leading ports in the central and south part of the Adriatic east coast. The leading ports are Dubrovnik, Korčula, Kotor, Split, Zadar and Šibenik. The horizontal axis indicates each year of the research, while the vertical axis shows number of cruise calls. Each port is marked with its colored bar, and on the top of each bar is a figure indicating the number of cruise calls of the designated port.

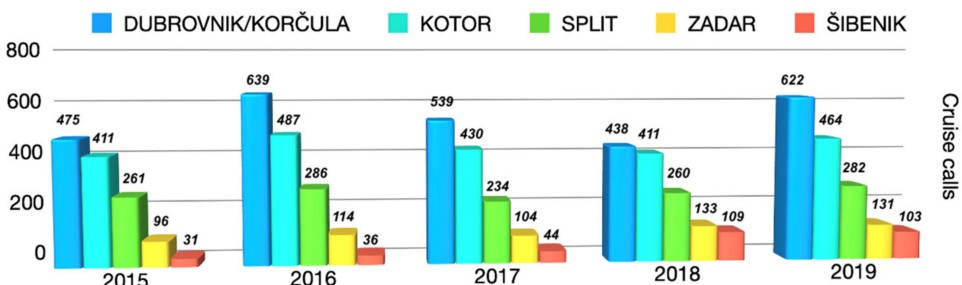

**Figure 4.** Total cruise calls from 2015 to 2019 in five leading ports in the central and south part of the Adriatic east coast.

All destinations in the central and south part of the Adriatic east coast measured growth in cruise calls, with slight oscillation in 2018. The total cruise calls increment in the central and south part of the Adriatic east coast from 2015 to 2019 was 25.75% compared to 18.70% in the Adriatic and only 4.30% in the Mediterranean region (Figure 4) [10].

Figure 5 shows total cruise passenger movement from 2015 to 2019 in five leading ports in the central and south part of the Adriatic east coast. The horizontal axis indicates each year of the research, while the vertical axis shows number of passenger movements in selected ports. Each port is marked with its coloured bar, and on the top of each bar is a figure indicating number of passenger movements in the selected port. Passenger movement trends follow the cruise ship call trends; the growth was constant with slight oscillation in 2018.

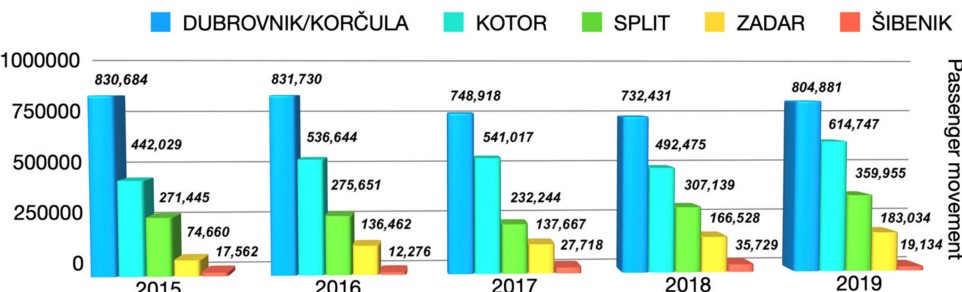

**Figure 5.** Total cruise passenger movement from 2015 to 2019 in five leading ports in the central and south part of the Adriatic east coast.

The number of total passenger movements in the central and south part of the Adriatic east coast increased by 46.07% in the period from 2015 to 2019 compared to 20.90% in the Adriatic and only 14.49% in the Mediterranean (Figure 4) [10].

Data on cruise ship calls and passenger movement show that the central and south part of the Adriatic east coast is the fastest growing region in the Adriatic Sea as well as in the Mediterranean.

### 1.2.1. Dubrovnik

Dubrovnik is the largest cruise port in Croatia and the second largest in the Adriatic, with 16% of cruise passenger traffic in the Adriatic [14]. The port of Dubrovnik is one of the most perspective Croatian ports and it has great potential to become one of Europe's leading cruise ports [16]. Dubrovnik is located in a prominent Adriatic cruise route, between Venice in the north and Kotor in the south. The growth of cruising tourism is very evident in the Croatian part of the Adriatic Sea, especially in Dubrovnik and Korčula [17]. Dubrovnik/Korčula had 475 cruise calls and 830,684 total cruise passenger movements in 2015 and 622 cruise calls and 804,881 total cruise passenger movements in 2019 (Figures 3 and 4). Due to the attractiveness and growing popularity of Dubrovnik Old Town on one side and its very limited area on the other, Dubrovnik has experienced the problem of over-tourism. A large number of cruise ships, cruising passengers and independent tourists at the same time causes congestion in Dubrovnik Old Town [16]. In order to solve the problem of congestion and to reduce navigational, environmental and social risks, cruising traffic have to be well planned and controlled. The Dubrovnik Port Authority has implemented measures which limited the number of cruise passengers up to 5000 per day and the number of cruise ships alongside up to two ships in passenger terminals and one ship at the anchorage at the same time [18]. The effects of the imposed measures are visible from the stable number of cruise calls and passenger movement trends in the period between 2015 and 2019 (Figures 3 and 4).

### 1.2.2. Korčula

Korčula is a relatively new destination in the cruising industry. In 2019, Korčula had 136 cruise calls and 35,957 cruise passengers (Figures 3 and 4). Although cruise traffic is still relatively low compared to the leading Adriatic cruise destinations, cruising trends are positive and have prospects for further growth [18]. Its geographical location near established cruise ports of Dubrovnik, Kotor and Split makes Korčula an ideal cruise transit port [19]. Korčula Old Town is one of the best examples of a fortified medieval town in the Mediterranean and it is listed in UNESCO's tentative list of outstanding world heritage sites [20]. Taking into consideration all the advantages of Korčula and future infrastructure investments in terminal Polačište, Korčula has a promising perspective as a desirable cruising destination [21].

### 1.2.3. Kotor

Kotor is the largest cruise destination in Montenegro and the third largest cruise port in the Adriatic Sea [1]. Kotor is located in Boka Kotorska Bay (fjord). The natural beauty and historical value of Kotor has been recognized by the UNESCO. In 2019, Kotor had 464 cruise calls and 614,747 cruise passengers' visits (Figures 3 and 4) [10]. Kotor can accommodate three to four cruise ships (up to 250 m length) and has a river berth for smaller cruise ships and three anchorages [22]. Investment in a modern cruise terminal and passenger facilities is required [23] in order to make port of Kotor safe in terms of navigational safety as well as in the terms of sustainability in the full meaning of the phrase sustainable development.

### 1.2.4. Split

The port of Split is the largest Croatian passenger port and the second largest Croatian cruise port after Dubrovnik [14]. Its geographical location in the vicinity of prominent

cruising destinations Dubrovnik and Kotor and rich historical heritage recognized by the UNESCO makes a Split competitive and perspective destination in the cruising market. Split has experienced remarkable touristic success and has evolved from a transit ferry destination to one of the leading touristic and cruising destinations in the Adriatic Sea. In 2019, the total passenger turnover was over 5.6 million passengers and 829,594 vehicles [24]. Split had 261 cruise calls and 251,455 total cruise passenger movements in 2015 and 282 cruise ship calls and 359,955 cruise passenger movements in 2019 (Figures 3 and 4) [10]. Cruise tourism in Split started in 2002, when 82 cruise ships with 20,616 passengers visited Split [24]. In the period between 2007 and 2016, cruise traffic almost tripled, while in the period between 2014 and 2016, it doubled in spite of the lack of cruise vessel berths and adequate passengers' facilities [14]. In 2017, the extension of the port operational quay and two external cruise berths with infrastructure (265 m long berth 26 and 245 m long berth 27) were completed as a part of a port development project. The new berths are capable to simultaneously accommodate two cruise ships of 320 and 270 m. The berths are equipped with border crossing points, sanitary facilities, an access road and other supporting infrastructure that offers high-quality services to passengers [14]. The extension of the quays and the construction of new external berths improved the maritime security and road traffic inside the port and have directly increased port service quality and competitiveness among other cruise ports in the region [25]. Split has the potential to become one of the Adriatic homeports, especially because the construction of a new terminal building has been included in short-term plans and better land access to the port is under consideration [25].

### 1.2.5. Zadar

Zadar is the second largest ferry port and the third largest cruise port in Croatia [14]. Zadar is located in the central part of the Adriatic east coast, and it is an important Croatian traffic node where continental traffic corridors meet the Adriatic Sea [16]. Its ideal location between notable cruise ships ports, Venice to the north and Split, Dubrovnik and Kotor to the south, makes Zadar an ideal cruise ship port option. Zadar has the potential to become an important destination in the cruise market due to its rich heritage and the vicinity of UNESCO sites and national parks but also due to its developed infrastructure, highway and railway connections and the vicinity of the international airport. In 2019, the total passenger turnover was 2,390,575 passengers and 484,690 vehicles [26]. Zadar had 96 cruise calls and 74,660 total cruise passenger movements in 2015, while in 2016, Zadar's cruising tourism exploded, with 114 cruise ship calls and 136,452 cruise passenger movements (Figures 3 and 4). In 2019, Zadar had 131 cruise ship calls and 183,034 cruise passenger movements (Figures 3 and 4) [10]. Zadar's cruise traffic and maritime traffic boom was caused by the completion of a new spacious port operation and handling area with four quays and a new passenger terminal in Gaženica. The new terminal can simultaneously accommodate seven ships in domestic traffic, two ships in international traffic and three cruise ships [26]. The construction of the new terminal gives Zadar a chance to become a homeport and a competitive cruise destination in the Adriatic region.

### 1.2.6. Šibenik

Šibenik is predominately a passenger ferry port with an approximate passenger turnover of 250,000 per year [27]. Šibenik is a relatively new cruising destination in the cruise market. The town is situated in the central Adriatic east coast near prominent cruising destinations such as Split, Dubrovnik and Kotor, and it has numerous national and natural parks in the vicinity. Šibenik had 31 cruise calls and 17,562 total cruise passenger movements in 2015 and 103 cruise ship calls and 19,134 cruise passenger movements in 2019 (Figures 3 and 4). The port of Šibenik is situated in a flooded estuary of the river Krka, and it is well sheltered from winds and waves. The entrance to the port is through St. Anthony channel, and only ships up to 50,000 GT and up to 260 m in length can enter the port. The port has two designated cruise ship berths that can simultaneously accommodate

two ships of 260 m length and 200 m length [27]. In order to make Šibenik an important cruise destination, investment in a modern passenger terminal with full cruise passenger service is required.

## 2. Materials and Methods

Research into cruise ships' navigational practices and cruise ships' trends in the central and south part of the Adriatic east coast is carried out since the region records the highest growth in cruise ship calls and passenger movement in the Mediterranean. Taking this and the fact that the strong expansion of cruise ship traffic has not been equally followed by the development of new navigational corridors and the implementation of improved protected measures into consideration, this makes the region ideal for cruise traffic monitoring and analysis.

The methodology of the research was based on the analysis of cruise industry trends in the Mediterranean region. The results brought the dynamic expansion of the central and south part of the Adriatic east coast to our attention. Strong cruise ship passenger movements and an increment in cruise ship calls in the region which has not had dense maritime traffic before offered valid grounds to identify newly established cruise ship routes as well as their influence on navigational and environmental safety. In addition, the research will determine cruise ships' traffic density in the vicinity of marine protected areas and will identify areas of high navigational and environmental risks. The obtained results and traffic comparison provide information regarding how often cruise ships use the Central Adriatic Separation Scheme. The methodology of the paper is presented in the diagram, in which all levels of the research are chronologically presented (Figure 6).

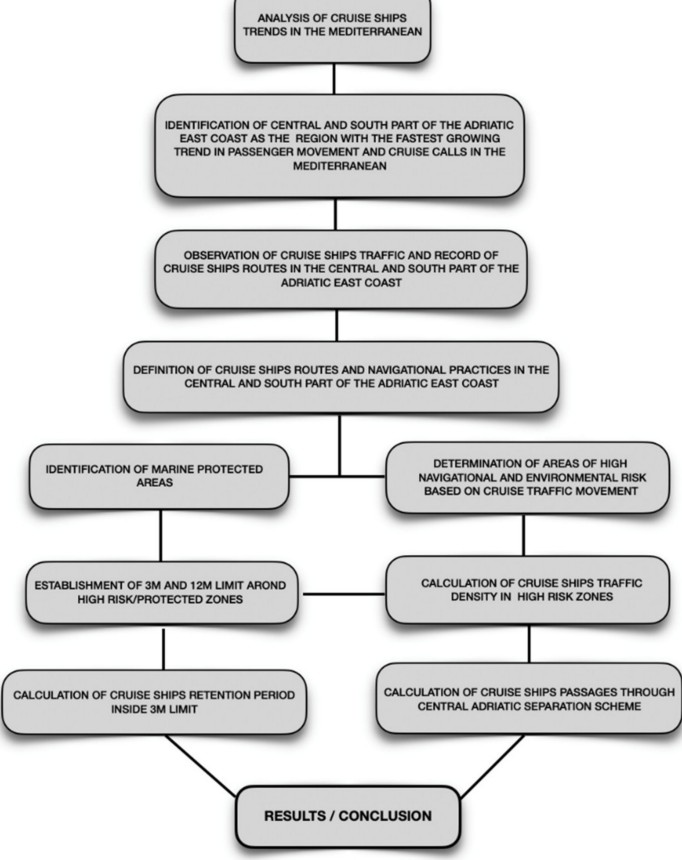

**Figure 6.** Research methodology diagram.

The analyses of leading cruise destinations trends are based on four-year (from 2015 to 2019) cruise passenger movement and cruise calls data. The data of cruise ships' movement (ships over 50,000 GT) were recorded on a daily basis in the period from August 2014 to

July 2015 using the Marine Traffic application. The data obtained from Perić (2016) [28] were processed and analyzed in the research. During the research, the routes of every cruise ship in the Adriatic east coast were recorded. The collection and analysis of all cruise routes offered real and detailed insight into cruise ships' navigational practices in the Adriatic east coast, in particular the central and south part of the Adriatic east coast.

### 2.1. Cruise Ships' Routes Presentation

Figure 7a–e show real cruise ship traffic movement in the researched period of time among leading cruise destinations in the east Adriatic region. Each figure represents the movement of one cruise ship. They are selected samples among all analyzed routes and represent general traffic flow in the east Adriatic coast. The presented figures are maps downloaded from the Marine Traffic application which are adjusted and edited with research data. Each figure shows different cruise ships' route patterns. Destinations are named, blue lines show the cruise ship route, black squares indicate separation schemes and the light blue line is the Croatian and Italian territorial waters border.

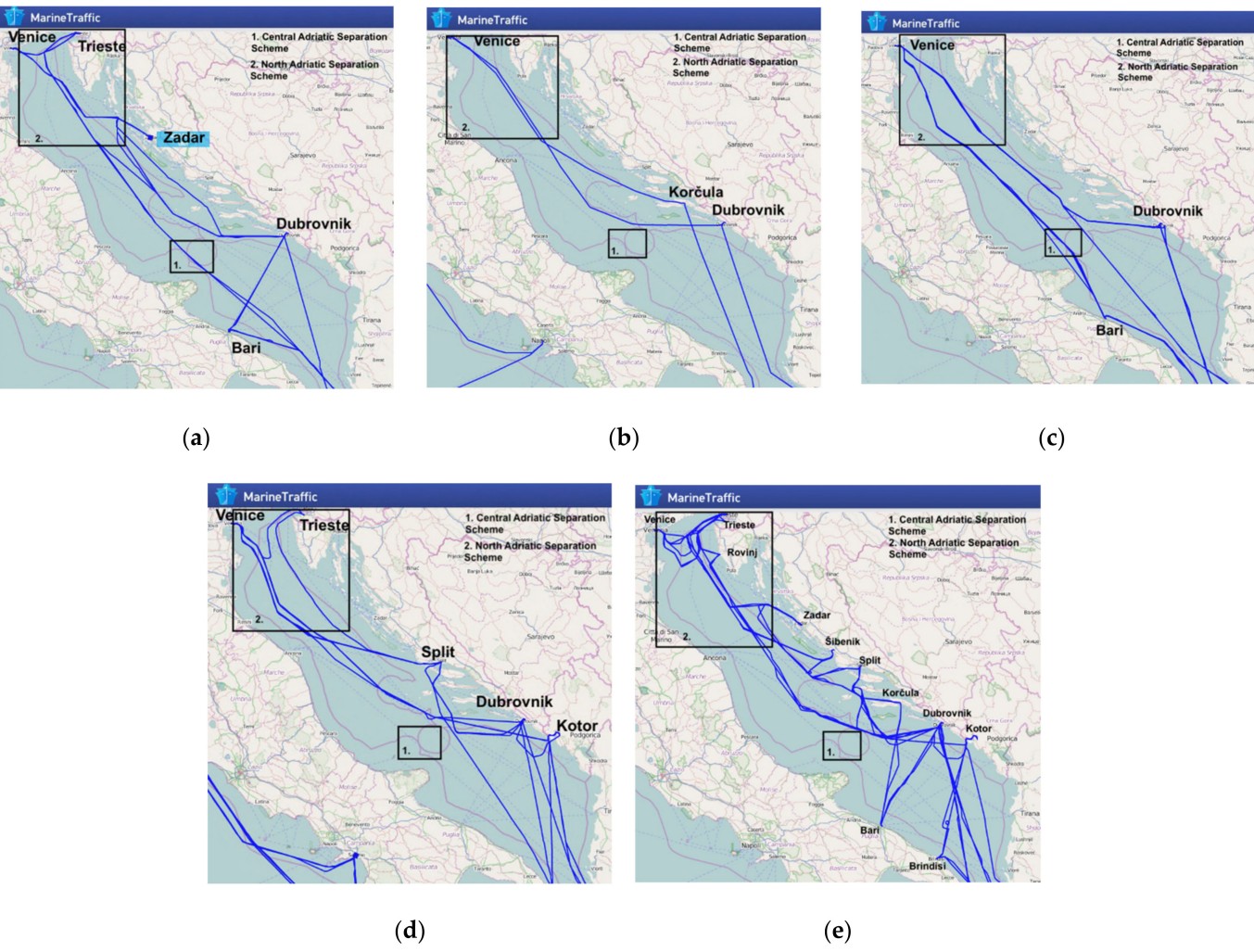

(a)　　　　　(b)　　　　　(c)

(d)　　　　　(e)

**Figure 7.** Real cruise ship traffic movement in the Adriatic east coast; (**a**) longitudinal cruise traffic corridor; (**b**) longitudinal cruise traffic corridor via port of Korčula; (**c**) international longitudinal cruise ships corridor through Croatian territorial waters in coastal navigation; (**d**) transversal cruise traffic corridor; (**e**) interaction between transversal and horizontal routes.

Figure 7a shows the routes of one selected cruise ship during a one-month period. The routes are part of two different circular itineraries. First itinerary: Venice—Trieste—Zadar—Dubrovnik—Bari—Mediterranean port—Venice. Second itinerary: Venice—Dubrovnik—Mediterranean ports—Venice. The figure emphasizes the longitudinal cruise traffic corridor

from the north Adriatic ports to the central and south Adriatic east coast ports and the corridor from the Mediterranean ports to the north Adriatic ports. Cruise ships heading from the north Adriatic ports to the central and southern Adriatic destinations, after leaving the North Adriatic Separation Scheme, stay in coastal navigation. They proceed along the other island ridge or in inland navigation heading to the south Adriatic regions. Cruise ships do not use the Central Adriatic Separation Scheme on the way to the south Adriatic east-coast destinations. The Central Adriatic Separation Scheme is only in use on direct routes, from Mediterranean ports or from the south Adriatic west-coast ports to the north Adriatic ports and vice versa.

Figure 7b shows the routes of one selected cruise ship's itinerary: Venice—Korčula—Mediterranean port—Dubrovnik—Venice. The figure emphasizes the longitudinal cruise traffic corridor from the north Adriatic ports to a Mediterranean port via the port of Korčula. Cruise ships follow the North Adriatic Separation Scheme, and when leaving the North Separation Scheme, cruise ships proceed in inland navigation through the Vis and Hvar channel to the port of Korčula. Departing Korčula cruise ships proceed through the channel between Lastovo and Mljet island heading to the Otranto strait.

Figure 7c shows the routes of one selected cruise ship during a one-month period. The routes are part of two different circular itineraries. First itinerary: Venice—Dubrovnik—Mediterranean port—Venice. Second itinerary: Venice—Bari—Mediterranean port—Venice. This figure emphasizes irregular navigational practice where cruise ships on the international longitudinal route from Mediterranean ports to the north Adriatic ports proceed in coastal navigation inside Croatian territorial waters. The cruise ship did not use the Central Adriatic Separation Scheme, and the route was not planned for the shortest stay inside Croatian territorial waters. On the contrary, the route proceeded in coastal navigation between the islands on the way to the north Adriatic Italian ports. The Central Adriatic Separation Scheme was in use only on itineraries from the south Adriatic west coast ports to the north Adriatic ports and vice versa.

Figure 7d shows the routes of one selected cruise ship during a two-month period. The routes are part of four different circular itineraries. First itinerary: Venice—Dubrovnik—Kotor—Mediterranean port—Venice. Second itinerary: Venice—Split—Mediterranean port—Venice. Third itinerary: Venice—Kotor—Mediterranean port—Venice. Forth itinerary: Venice—Dubrovnik—Mediterranean port—Venice. The figure emphasizes the transversal navigational corridor that passes the west coast of Sušac island and between Sušac island and Lastovo island.

Figure 7e shows the routes of one selected cruise ship during a two-month period. The routes are part of different itineraries among: Venice—Trieste—Rovinj—Zadar—Šibenik—Split—Korčula—Dubrovnik—Kotor—Brindisi—Bari. The figure emphasizes the interaction between horizontal and vertical routes. It shows vertical routes that pass west of Sušac island and between Sušac and Lastovo island and routes between Lastovo island and Mljet island.

*2.2. Calculation of Traffic Density and Distance Cruise Ships Pass from the Island Shores*

The calculation of traffic density and the distance cruise ships pass from the island shores is based on traffic analysis during the busiest five-month period (August–October 2014 and June–July 2015). In order to identify density and determine the distance cruise ships traffic pass from the island shores and marine protected areas, regions of high navigational and environmental risk were selected. Regions of high navigational and environmental risk were selected based on the traffic monitoring results, geographical particularities, and environmental sensitivities of the selected area. Regions of high interest were Svetac, Biševo and Sušac and Lastovo islands, as well as the Biševo–Vis passage, Vis channel, Hvar channel and the Sušac–Lastovo and Lastovo–Mljet passages.

On the chart, around the selected islands, 3 M diameter black circles are set. A distance of 3 M was set as a safe navigational reference and was in reference to International Convention for the Prevention of Pollution from Ships Annex IV (MARPOL), which stipulates

a 3 M limit for ships' sanitary water (grey water) discharge. The second reference was the 12 M territorial water limit marked with a light blue line. The distance of 12 M was also chosen in reference to International Convention for the Prevention of Pollution from Ships Annex IV (MARPOL), which stipulates a 12 M limit for ships' untreated wastewater (black water) discharge. Protected areas on the chart are marked with a green line (Figure 8a–c).

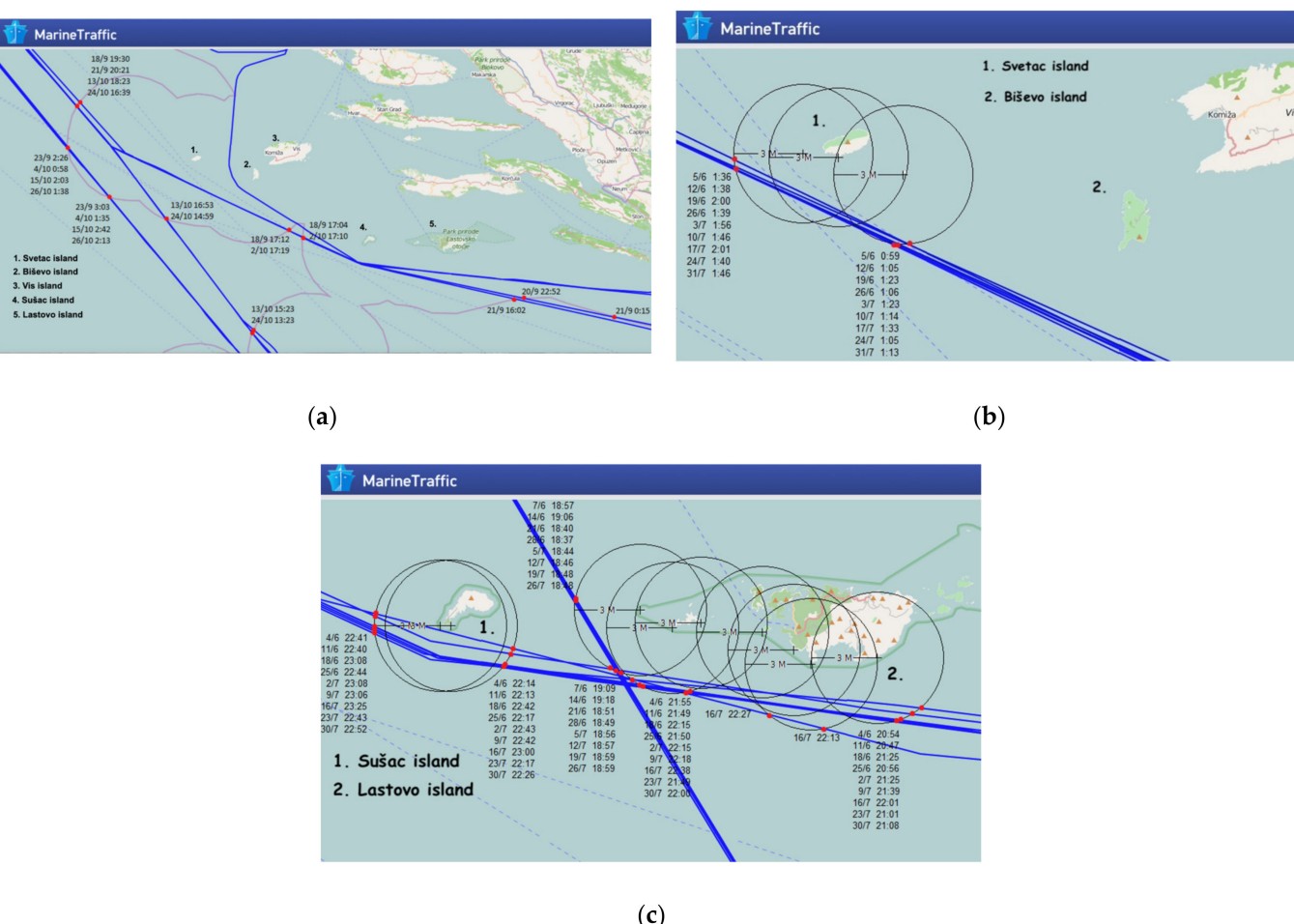

**(a)**                    **(b)**

**(c)**

**Figure 8.** (**a**) Presentation of cruise traffic distance from the islands' shores and display of >12 M retention calculation in the central and south Adriatic east coast; (**b**) presentation of cruise traffic distance from the islands' shore and display of >3 M retention calculation with focus on the islands of Svetac and Biševo; (**c**) presentation of cruise traffic distance from the islands' shores and display of >3 M retention calculation with focus on the islands of Sušac and Lastovo.

Figure 8a–c are routing samples in the central and south Adriatic east coast, selected among all monitored cruise ship routes. Each of them represents the movement of one cruise ship in the researched period of time. Areas of interest are numbered and named, the blue lines show the cruise ship routes, the black circles indicate 3 M distance from the shore, dates and times indicate when a ship entered and left a 3 M or 12 M limit and the light blue lines show the Croatian territorial waters border (12 M limit). Analyzing Figure 8a–c and observing the times and dates of ships' arrivals and departures from 3 M or 12 M limits, it is evident that cruise routes are repetitive and have become navigational routine in the central and south part of the Adriatic east coast.Cruise ships' density in the researched areas was calculated following each cruise ship's route. In the process, the time of each ship's entrance and exit to 3 M and 12 M limits was recorded. Pre-set 3 M diameter circles and 12 M territorial waters limits were used as the main references in Figure 8a–c. Cruise ships that crossed the 3 M circle limit were considered to pass inside 3 M from the shore, cruise ship routes which passed close to the 3 M circle limit were considered to pass with a distance 3 M to 6 M from the shore, and open sea routes closer to the 12 M territorial water

border were considered to pass more than 6 M from the shore. Each cruise ship was counted and put into one of three categories according to the distance they passed from the islands shore: (1) <3 M from shore, (2) 3 M to 6 M from shore and (3) >6 M from shore. Cruise ships retention period inside 3 M from shore is also calculated. The time difference between the entrance and exit from the 3 M diameter was taken into consideration, and it was calculated and expressed in minutes. The collected data related to the distance cruise ships pass from the islands' shores and traffic density are grouped in four tables (Tables A1–A4). Each table is related to one high-risk navigational and environmental region. Information is given for each month (August–October 2014 and June–July 2015) and cumulatively. Data available from each table are the number of cruise ships that pass <3 M from shore, 3–6 M from shore and >6 M from shore, the retention period <3 M from shore expressed in minutes and the average retention period expressed in minutes. Cumulative data are expressed as totals and give a summary of cruise ships' impact on the designated area.

The number of cruise ship transits through Svetac—Biševo, Sušac—Lastovo and Lastovo–Mljet passages were calculated separately, as well as number of transits through the Central Adriatic Separation Scheme (Tables A5–A8). Information is given for each month of the research and cumulatively. The comparison between routes that used the Central Adriatic Separation Scheme and routes that kept north of the Central Adriatic Separation Scheme offers an important conclusion regarding cruise ships' navigational practice in the central and south Adriatic east coast. The number of cruise ships that used the Central Adriatic Separation Scheme were counted from cruise traffic monitoring. In order to make a comparison, the number of routes that passed north of the Central Adriatic Separation Scheme were calculated. Sušac island was taken as a reference; it is the island located north of the Central Adriatic Separation Scheme, it is positioned on the southern point of the longitudinal corridor that cruise ships frequently use, and it is the closest land from the Central Adriatic Separation Scheme. Palagruža island, being a remote island at the immediate border to the Central Adriatic Scheme, does not offer reliable data and was not taken as the reference.

The calculation was carried out as follows: From Table A2, cumulative information related to the number of cruise ships that passed 3–6 M (265 cruise ships) and <6 M (30 cruise ships) from Sušac island were added (data < 3 M were not taken into consideration, as cruise ships that passed <3 M from the shore passed 3 M–6 M from the shore too). From the result, the number of vertical routes that transited through the Sušac–Lastovo passage (58) were subtracted (Table A6), because for the comparison, only horizontal routes had to be taken in consideration. The final result was: 237 cruise ships kept north of the Central Adriatic Separation Scheme passing in the vicinity of Sušac island, while 187 cruise ships used the Central Adriatic Separation Scheme.

## 3. Results and Discussion

Cruise traffic expansion in the central and south part of the Adriatic east coast has established new frequently used routes as navigational options to the Central Adriatic Separation Scheme. New insight into cruise ships' navigational practices was obtained by processing and analyzing data from a one-year survey of cruise ships in the Adriatic Sea from Perić (2016). The data acquired are charted in two integrated charts (Figure 9a,b). Figure 7a–e justify the data in Figure 9a,b.

Figure 9a defines the longitudinal and transversal traffic flow of cruise ships over 50,000 GT in the Adriatic east coast. The figure shows cruise ships' routes (black lines), the well-defined North Adriatic Separation Scheme (red square 1) and the shorter, less defined Central Adriatic Separation Scheme (red square 2). The North Adriatic Separation Scheme directs the cruise traffic on steady routes without any oscillation, while the Central Adriatic Separation Scheme offers more routing options, and therefore, cruise routes are more dispersed than in the north Adriatic.

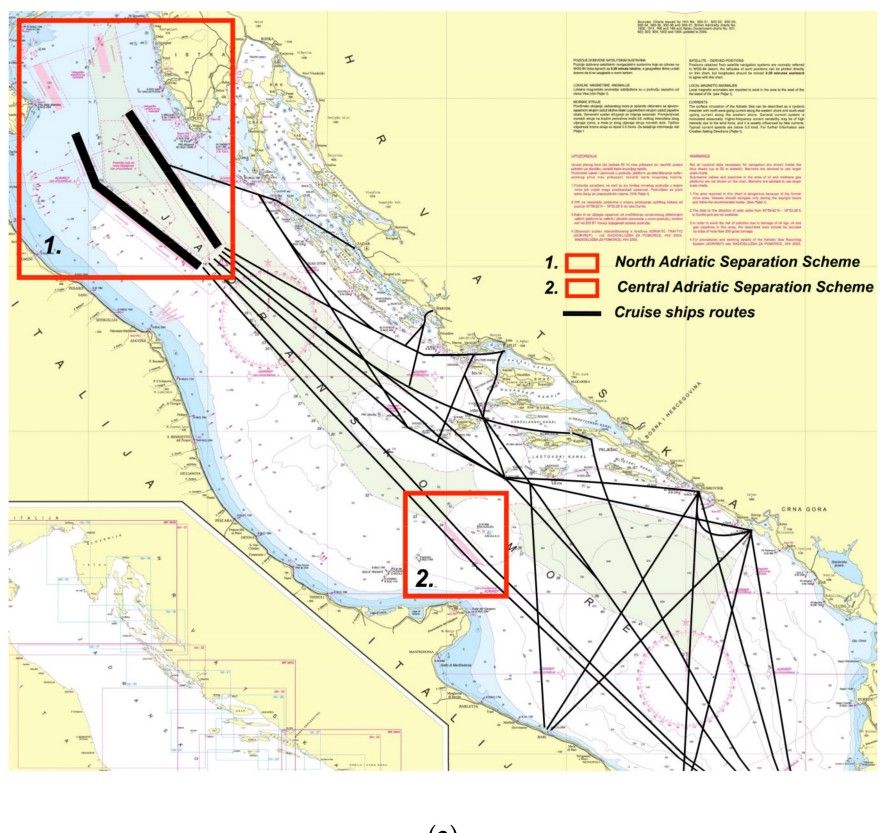

(**a**)

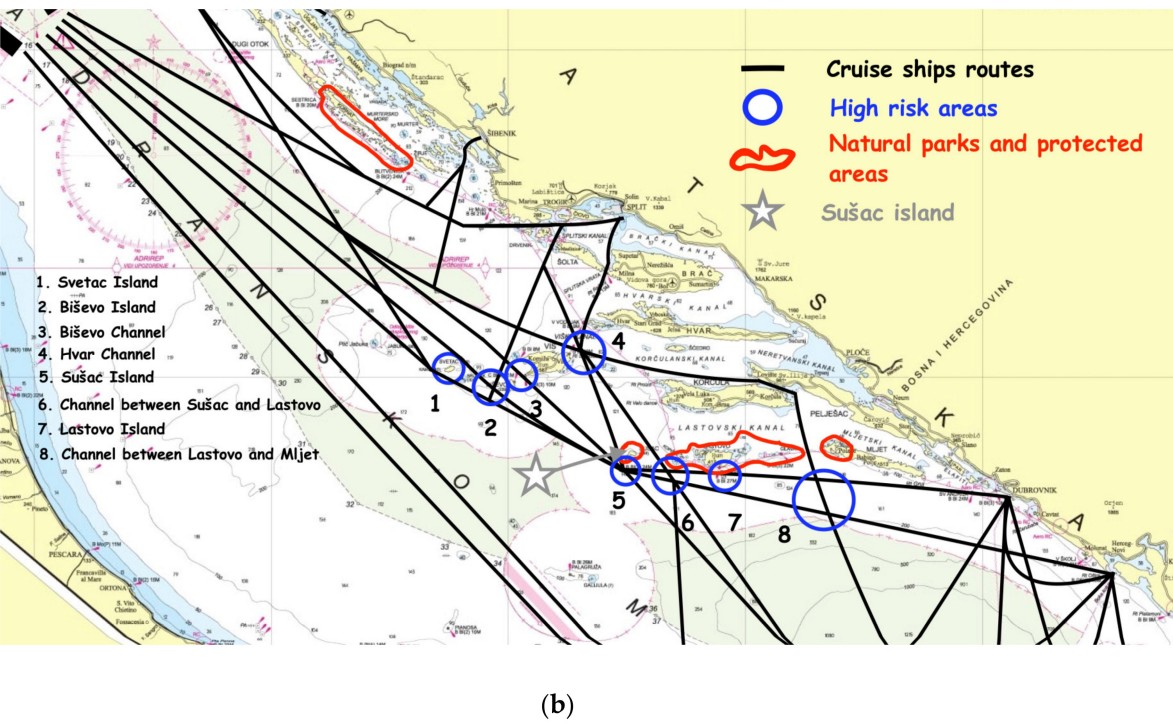

(**b**)

**Figure 9.** (**a**) Cruise traffic in the Adriatic east coast [7]. Black lines—cruise ships' routes; red squares—red square 1: the North Adriatic Separation Scheme; red square 2: the Central Adriatic Separation Scheme. (**b**) Cruise ships' routes and high-risk zones in the central and south part of the east Adriatic. Black lines—cruise ships routes; red squares—red square 1: the North Adriatic Separation Scheme; red square 2: the Central Adriatic Separation Scheme; red curves—marine protected areas; grey star—the island of Sušac; blue circles—areas of high navigational risks.

According to the traffic monitoring and route analysis in the east Adriatic, cruise ships' movements can be defined in three standard patterns:

1.  From the north Adriatic ports to the central and south destination of the Adriatic east coast. Cruise ships departing from the north Adriatic ports proceed on southeasterly courses using the North Adriatic Separation Scheme. Once they leave the North Adriatic Separation Scheme, they follow the route along the outer skirts of the islands along the east Adriatic coast. They do not use the Central Adriatic Separation Scheme (Figure 7a–e).

2.  From the north Adriatic ports to the Otranto strait and vice versa. Cruise ships departing from the north Adriatic ports proceed on southeasterly courses using the North Adriatic and Central Adriatic Separation scheme, heading to the Otranto strait. On northwesterly courses, cruise ships proceed from the Otranto strait to the north Adriatic ports using the Central and North Adriatic Separation Scheme (Figure 7a,c).

3.  From the Otranto strait to the northern Adriatic via south Adriatic east coast destinations. Cruise ships arriving from Mediterranean ports through the Otranto strait proceed directly to the south Adriatic east coast destinations. Departing from the South Adriatic east coast destinations, cruise ships proceed in coastal navigation along the outer skirts of islands and keep out of the Central Adriatic Separation Scheme. In northwesterly courses, they join the North Adriatic Separation Scheme on the way to north Adriatic ports (Figure 7a–e).

Cruise ships' traffic monitoring has shown irregular navigational practice where cruise ships on international voyages in transit through Croatian territorial waters (on routes: Montenegro—Italy or Mediterranean port to Italy) do not use the Central Adriatic Separation Scheme, nor are the routes planned to enable the shortest stays inside Croatian territorial waters. Cruise ships tend to choose coastal navigation between the islands in Croatian territorial waters until they reach the North Adriatic Separation Scheme (Figure 7c). The Central Adriatic Separation Scheme is only used on direct voyages from the north Adriatic ports to the Otranto strait and from the Otranto strait to north Adriatic ports.

*3.1. High-Risk Zones Created by Cruise Ships' Routes*

Figure 9b is a fragment of Figure 9a; it shows high-risk zones created by cruise ships' routes. Figure 9b shows cruise ships' traffic in the central and south part of the Adriatic east coast (black lines), the position of four marine protected areas (red curves) and eight areas of high navigational and environmental risks (blue circles). Sušac island was selected as a reference point being the closest to the Central Adriatic Separation Scheme (grey star). Four marine protected areas are Kornati National Park, Mljet National Park, Lastovo Nature Park and Sušac Island Nature Park. Eight areas of high navigational and environmental risks were selected based on cruise ships' routes data (the vicinity that cruise ship routes pass from the islands' shores and marine protected areas and cruise routes' mutual interaction). Designated areas of high navigational risks are the islands of Svetac, Biševo, Sušac and Lastovo, as well as the Svetac–Biševo passage, Hvar channel and the Sušac–Lastovo and Lastovo–Mljet passages.

The comparison of cruise traffic density in the Central Adriatic Separation Scheme and cruise traffic density in the vicinity of the island of Sušac provide valuable information on cruise ships' navigational principles in newly established and navigationally less defined cruise regions. The research showed that cruise ships used the Central Adriatic Separation Scheme 187 times, while cruise ships kept north of the Central Adriatic Separation Scheme and passed south of Sušac island 237 times.

Figure 9b shows that cruise ships frequently pass through areas of high navigational and environmental risk which are geographically restricted, navigationally challenging and environmentally sensitive areas. Cruise ships' routes that pass very close to outer island shores as well as between the islands have become standard navigational practice for cruise ships. Additionally, longitudinal and transversal routes' interaction in coastal navigation

puts cruise ships in crossing, head on and overtaking situations, which increase the risk of collision grounding and pollution, especially in areas of high navigational risks.

During the research period, cruise ships' traffic inside designated high-risk areas was thoroughly analyzed. The results are presented for each high-risk zone separately.

### 3.1.1. Svetac Island

Svetac island is part of the northwesterly and southeasterly longitudinal corridor. Cruise ships departing from south Adriatic east-coast destinations to the northern Adriatic ports on northwesterly routes do not use the Central Adriatic Separation Scheme. The Central Adriatic Separation Scheme is also not in used in southeasterly routes from the northern Adriatic ports to the south Adriatic east-coast destinations. Cruise ships proceed along the outer islands route, passing south or north of Svetac island's shores.

Traffic monitoring showed that 76 cruise ships passed less than 3 M from the shores of Svetac island with average retention periods of 16.1 min. A total of 155 cruise ships passed with 3 M to 6 M distances, and 47 cruise ships kept 6 M or more from the island's shores (Table A3). Traffic monitoring showed that two cruise ships met less than 3 M from the island shores six times in a 90 min period. On one occasion, during a 90 min period, three cruise ships met less than 3 M from the island's shore.

### 3.1.2. Biševo Island

Biševo island is part of the longitudinal and transversal corridor that cruise ships use. Longitudinal southeasterly routes proceed to the south Adriatic east-coast destinations, while northwesterly routes head to north and central Adriatic ports. Transversal routes in northerly and southerly directions connect the port of Split with southern Adriatic destinations and the Italian coast.

During the research period, 21 cruise ships passed less than 3 M from Biševo island with average retention periods of 19.23 min, 51 cruise ships passed with 3 M to 6 M distances from Biševo island, 6 cruise ships kept 6 M and more from Biševo island and 43 cruise ships passed through the Svetac and Biševo passage (Table A4). Traffic monitoring showed that two cruise ships met less than 3 M from Biševo island's shores five times in a 90 min period.

### 3.1.3. Biševo Channel

Cruise ships' routes through the Biševo channel are not usual; however, despite restrictions, cruise ships' passage through the Biševo channel was detected. During the monitoring period, cruise ships passed through the Biševo channel five times with average retention periods of 25.2 min less than 3 M from the shore. The passage was carried out by identical cruise ships, which shows that the route has become a standard navigational routine for monitored cruise ships. The observed cruise route puts cruise ships in immediate grounding and collision danger.

### 3.1.4. Hvar Channel

Hvar channel is situated between the island of Hvar and Vis in the central Adriatic region. It is located in an intersection of the longitudinal and transversal routes. The longitudinal inland cruising route connects the northern and central Adriatic region with southern Adriatic ports. The transversal route connects Split with the south Adriatic coast and Italy.

The main navigational risk is the crossing of transversal and longitudinal routes and head on situations in the restricted area of Hvar channel. During the research period, longitudinal inland cruise ships' traffic was not constant; on the other hand, transversal traffic was frequent. In addition to cruise traffic, there was a frequent ferry and catamaran connection from Vis and Lastovo islands to Split and vice versa. In addition to that, the area of Hvar channel is touristy and very popular with developed nautical tourism and

dense leisure craft traffic. Taking the above into consideration, navigation in Hvar channel has to be taken with precaution, since the risk of collision and grounding is elevated.

### 3.1.5. Sušac Island

Longitudinal northwesterly and southwesterly routes as well as transversal southerly and northerly routes often pass along the coast of Sušac island. Cruise ships on northwesterly and southeasterly longitudinal routes connecting south Adriatic east-coast destinations and north Adriatic destinations (and vice versa) do not use the Central Adriatic Separation Scheme. They proceed along outer island routes passing the south shores of Sušac island. Transversal routes pass the eastern and western shores of Sušac island. Cruise ships on southern transversal routes depart from northern Adriatic ports or the port of Split and proceed to the western Adriatic coast or join the longitudinal corridor on the way to Dubrovink, Kotor or the Otranto strait. Northerly transversal routes connect western Adriatic ports, the Otranto strait, Dubrovnik or Kotor to central Adriatic ports or northern Adriatic ports.

During the research period, 101 cruise ships passed less than 3 M from Sušac island with average retention periods of 13.84 min, 265 cruise ships passed with 3 M to 6 M distances from the south shores of Lastovo, while 30 cruise ships kept distances of 6 M or more from the south island shores (Table A2). Traffic monitoring showed that two cruise ships met less than 3 M from the island shores seventeen times in a 90 min period.

### 3.1.6. Sušac–Lastovo Passage

The passage between Sušac island and Lastovo island is part of a transversal route that cruise ships use on northerly and southerly courses. Cruise ships on southerly courses proceed through the Sušac–Lastovo passage to the western Adriatic coast or they join the longitudinal corridor on the way to Dubrovnik, Kotor or the Otranto strait. Meanwhile, cruise ships on northerly courses proceed through the Sušac–Lastovo passage from western Adriatic shores or from the longitudinal corridor from Dubrovnik or Kotor on the way to the central and north Adriatic ports.

During the research period, 58 cruise ships passed through the Sušac–Lastovo passage (Table A6). Traffic monitoring showed that two cruise ships met in the passage within a 3 M distance from the shore six times in a 90 min period. On one occasion, four cruise ships met inside the passage within 3 M from the shore in a 90 min period.

### 3.1.7. Lastovo Island

Longitudinal northwesterly and southeasterly cruise ship routes often pass along the south coast of Lastovo island. Cruise ships on a northeasterly longitudinal route from Kotor and Dubrovnik to northern Adriatic ports do not use the Central Adriatic Separation Scheme. They proceed in the east coast outer island route, heading along the south coast of Lastovo island and Sušac island until they reach the Northern Adriatic Separation Scheme. Cruise ships on southeasterly routes from the north Adriatic ports, after leaving the Northern Adriatic Separation Scheme, proceed in east coast outer island route along Lastovo south coast to Dubrovnik and Kotor.

During the research period, 113 cruise ships passed less than 3 M from the south shores of the island with average retention periods of 21.2 min, 281 cruise ships passed with 3 M to 6 M distances from the south shores of Lastovo, while 19 cruise ships kept distances of 6 M or more from the south island shores (Table A1). Traffic monitoring showed that two cruise ships met within 3 M from the shore six times in a 90 min period.

### 3.1.8. Lastovo–Mljet Passage

The passage between Lastovo island and Mljet island is precisely between Glavat island and Mljet island, which cruise ships use on northerly courses and on southerly courses when arriving/departing to or from Korčula island. During the summer season,

there is dense sailing boat, yacht and leisure boat traffic, which often interacts with cruise ships' transit through the passage.

During the research, 30 ships passed through the Lastovo–Mljet passage (Table A7). Traffic monitoring showed that two cruise ships met in the passage within a 3 M distance from the shore two times in a 90 min period. On six occasions, two cruise ships met in a crossing course 3 M to 6 M from the shore in a 90 min period.

The results of the research give a new conclusion and new information from the paper 'Main sailing routes in the Adriatic' [10], in which one of conclusions states that maritime flow in the central Adriatic region is mostly directed through the Central Adriatic Separation Scheme and that maritime accidents are rare, which indicates good maritime coordination. Additionally, in a statement in the paper 'Analysis of the maritime traffic in central part of the Adriatic' [9], one of the conclusions states that the greater part of the longitudinal sailing route extends in the area of sufficient depth and width where there is no significant danger to navigation with the exception of the danger of collision with opposite and transverse traffic and the danger of grounding in the broader area of Palagruža island.

The study brought elevated navigational, safety and environmental risks in the navigationally less defined region of the central and south Adriatic east coast to our attention. The present cruise routing practice presents serious environmental and safety risks to protected areas of the Lastovo Natural Park, Sušac island and the Mljet National Park because the preserved natural ambience and ecosystems give these locations high natural, national and international value.

The analysis of cruise ship movement shows that high-risk routes are repetitive and have become navigational standard routine for cruise ships. This practice questions efficient maritime coordination in the researched area. It is of high importance that cruise traffic expansion is well controlled and equally complemented with investment, the development of routing systems, the implementation of restricted areas, efficient traffic control and straightened maritime regulations in order to ensure the sustainable development of the central and south Adriatic east coast.

This research showed the standard cruise shipping navigational practices in coastal navigation in areas that have been recently discovered by the cruise industry. The observed cruise ship routing practice can be related to any costal region where strong cruise traffic expansion has not been equally supported by investment, the development of routing systems, the implementation of restricted areas and efficient marine traffic coordination. The results are not only related to the central and south part of the Adriatic east coast; on the contrary, they show cruise ships' navigational practice that can be associated globally to any developing cruise region.

## 4. Limitations of the Study

The monitoring of cruise ships' traffic was carried out in period from August 2014 to June 2015, and although the period of the research does not display the current status, the results showed cruise ships' navigational routines where selected cruise routes are constant as cruise ships operate on circular itineraries. The defined navigational practice in costal navigation has become standard operational procedure for cruise ships. The results are not closely related to the time period; on the contrary, they reveal established cruise ships' navigational practice in coastal navigation.

Cruise industry trends were analyzed for the period from 2015 to 2019. Further research was not possible due to the COVID pandemic and the subsequent inactivity in the cruise ship industry. However, the obtained results directed research to the central and south part of the Adriatic east coast. The uniqueness of the region gave solid ground for the detailed analysis of cruise ships' routing practice in coastal navigation. Global cruise industry trends and cruise ships' routing practices in costal navigation raised the question of the sustainable development of coastal regions that have not been involved in cruise tourism before.

The research covered sizes of cruise ships from 50,000 GT and more. The size limit was placed in order to only take large cruise ships with higher environmental impacts and elevated safety risks into consideration. However, smaller cruise ships under 50,000 GT should be taken into consideration in future research, as they are also important factors of navigational safety and the sustainable development of coastal areas.

## 5. Conclusions

Research on leading cruise destination trends in the Adriatic region for the period from 2015 to 2019 showed that the central and south part of the Adriatic east coast is the region with the highest growth in cruise ship calls and passenger movement in the Adriatic and in the entire Mediterranean region. The strong expansion of cruise tourism has created an impact on sustainable development in already established and popular destinations in the Adriatic region such as Venice and Dubrovnik. As a result of imposed restrictions in Venice and Dubrovnik and the development of the cruise industry in general, the cruise industry has discovered new destinations in the Adriatic. The central and south part of the Adriatic east coast has taken advantage of the present circumstances and became an important factor in the Mediterranean cruise market. Destinations such as Zadar, Šibenik, Split, Korčula and Kotor together with established Dubrovnik have become recognizable cruise ports with solid perspectives for continuing growth.

New cruise destinations in the central and south part of the Adriatic east coast have created new cruise ship routes in a region that did not have dense maritime traffic before. In addition, the region is of important national and natural value due to its unspoiled beauty and natural protected areas. The aim of the paper was to define individually implemented cruise ship routing practices in costal navigation, to determine cruise ships' traffic density in the vicinity of maritime protected areas and to identify areas of high safety, navigational and environmental risk.

The results of this research show that cruise ships do not use the Central Adriatic Separation Scheme on the way to the south Adriatic ports; they keep north of the Central Adriatic Separation Scheme and proceed in coastal navigation along outer island shores. Navigation close to the outer island shores along outer island ridges as well as between the islands has become standard navigational practice for cruise ships. Additionally, longitudinal and transversal routes' interaction in coastal navigation puts cruise ships in crossing, head on and overtaking situations, which increases the risk of collision, grounding and pollution, especially in environmentally sensitive, naturally preserved, navigationally challenging and restricted areas. The execution of high-risk cruise ship routes, which have become standard navigational practice, shows a lack of maritime regulation and coordination. The Adriatic region has not had major maritime accidents recently; however, with the present navigational practice, maritime accidents with serious safety, environmental and economic consequences can easily occur.

The study has brought to our attention how efficient traffic coordination is of high importance for regulated and safe maritime traffic. With that in mind, it is of high importance that cruise traffic expansion is well controlled and equally complemented with investment and the implementation and development of routing systems, efficient traffic control and maritime regulation.

The results of the study gave perspectives for further research into cruise ships' practices in coastal navigation with a focus on cruise ships' environmental impact and preventive measures. Cruise ships' practices under 50,000 GT and their impact on coastal areas should be also taken into consideration in future research. In addition to environmental impacts, the results of the study offered valid grounds to plan the modification and extension of the Central Adriatic Separation Scheme and the implementation of restricted areas around high-risk and marine protected areas of the central and south part of the Adriatic east coast.

The obtained results offer a general overview of cruise ships' navigational practices in coastal navigation not only in the central and south Adriatic east coast region but in any coastal region in the world.

**Author Contributions:** Conceptualization, J.D.; methodology, J.D.; validation, J.D.; formal analysis, J.D.; investigation, J.D.; resources, J.D. and T.P.; data curation, T.P. and J.D.; writing—original draft preparation, J.D.; writing—review and editing, J.D., T.P. and G.J.M.; visualization, J.D.; supervision, J.D., T.P. and G.J.M. All authors have read and agreed to the published version of the manuscript.

**Funding:** This research received no external funding.

**Institutional Review Board Statement:** The study did not require ethical approval, nor was involved in animal, human studies.

**Informed Consent Statement:** The study is not involving humans.

**Data Availability Statement:** Data used for the analyses will be available upon request from the corresponding author.

**Conflicts of Interest:** The authors declare no conflict of interest.

## Abbreviations

| | |
|---|---|
| GDP | Gross Domestic Product |
| IMO | International Maritime Organisation |
| COVID | Coronavirus disease |
| UNESCO | The United Nations Educational, Scientific and Cultural Organization |
| GT | Gross Tonnage |
| M | Nautical mile (1852 m) |
| MARPOL | International Convention for the Prevention of Pollution from Ships |

## Appendix A

**Table A1.** Lastovo island—density, distance and retention period of cruise ships' traffic in selected high-risk navigational and environmental region.

| Month | <3 M | 3–6 M | >6 M | Retention Period <3 M (min) | Average (min) |
|---|---|---|---|---|---|
| August 2014 | 17 | 49 | 1 | 319 | 18.8 |
| September 2014 | 20 | 64 | 8 | 356 | 17.8 |
| October 2014 | 16 | 39 | 7 | 269 | 16.8 |
| June 2015 | 31 | 61 | 3 | 712 | 23.0 |
| July 2015 | 29 | 68 | 0 | 854 | 29.4 |
| Total | 113 | 281 | 19 | 2510 | 21.16 |

**Table A2.** Sušac island—density, distance and retention period of cruise ships' traffic in selected high-risk navigational and environmental region.

| Month | <3 M | 3–6 M | >6 M | Retention Period <3 M (min) | Average (min) |
|---|---|---|---|---|---|
| August 2014 | 15 | 38 | 4 | 200 | 13.3 |
| September 2014 | 17 | 60 | 9 | 140 | 8.2 |
| October 2014 | 19 | 41 | 7 | 243 | 12.8 |
| June 2015 | 26 | 60 | 7 | 451 | 17.3 |
| July 2015 | 24 | 66 | 3 | 423 | 17.6 |
| Total | 101 | 265 | 30 | 1457 | 13.84 |

**Table A3.** Svetac island—density, distance and retention period of cruise ships' traffic in selected high-risk navigational and environmental region.

| Month | <3 M | 3–6 M | >6 M | Retention Period <3 M (min) | Average (min) |
|---|---|---|---|---|---|
| August 2014 | 13 | 26 | 3 | 255 | 17.3 |
| September 2014 | 15 | 35 | 18 | 143 | 9.5 |
| October 2014 | 17 | 25 | 8 | 233 | 13.7 |
| June 2015 | 18 | 40 | 7 | 369 | 20.5 |
| July 2015 | 13 | 29 | 11 | 745 | 19.7 |
| Total | 76 | 155 | 30 | 1745 | 16.14 |

**Table A4.** Biševo island—density, distance and retention period of cruise ships' traffic in selected high-risk navigational and environmental region.

| Month | <3 M | 3–6 M | >6 M | Retention Period <3 M (min) | Average (min) |
|---|---|---|---|---|---|
| August 2014 | 10 | 11 | 1 | 156 | 15.6 |
| September 2014 | 4 | 15 | 1 | 66 | 16.5 |
| October 2014 | 3 | 9 | 0 | 61 | 20.3 |
| June 2015 | 0 | 6 | 1 | 0 | 0 |
| July 2015 | 4 | 10 | 3 | 98 | 24.5 |
| Total | 21 | 51 | 6 | 381 | 19.23 |

**Table A5.** Svetac–Biševo passage—number of cruise ships' transits in selected high-risk navigational and environmental locations (passages).

| Month | Number of Transits |
|---|---|
| August 2014 | 8 |
| September 2014 | 14 |
| October 2014 | 6 |
| June 2015 | 6 |
| July 2015 | 9 |
| Total | 43 |

**Table A6.** Sušac–Lastovo passage—number of cruise ships' transits in selected high-risk navigational and environmental locations (passages).

| Month | Number of Transits |
|---|---|
| August 2014 | 5 |
| September 2014 | 14 |
| October 2014 | 7 |
| June 2015 | 13 |
| July 2015 | 19 |
| Total | 58 |

**Table A7.** Lastovo–Mljet passage—number of cruise ships' transits in selected high-risk navigational and environmental locations (passages).

| Month | Number of Transits |
|---|---|
| August 2014 | 7 |
| September 2014 | 9 |
| October 2014 | 2 |
| June 2015 | 9 |
| July 2015 | 3 |
| Total | 30 |

**Table A8.** Central Adriatic Separation Scheme—number of cruise ships' transits.

| Month | Number of Transits |
|---|---|
| August 2014 | 24 |
| September 2014 | 42 |
| October 2014 | 37 |
| June 2015 | 42 |
| July 2015 | 42 |
| Total | 187 |

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
