# Peer review of "Cruise Industry Trends and Cruise Ships’ Navigational Practices in the Central and South Part of the Adriatic East Coast Affecting Navigational Safety and Sustainable Development"

_applsci, doi:10.3390/app12146884_

Round 1

Author Response

Good afternoon Mrs. or Mr.

Thank you for your review and feed back received. Please find attached corrections and responses to your reviews.

Best Regards

Josip Dorigatti

Reviewer 2 Report

A well prepared research and paper.

Nevertheless, few general doubts to be cleared by the Authors:

1. Considering current date of review (2022) why is the last year of the research 2019? And Cruise lines finish on July 2015? A long time ago and a real risk to have undated tables and results. Any kind of justification for that period analysed should be clarified.

2. No section Discussion maks the paper less valuable for the future research. My recommendation is to extent the paper by this section.

3. The title says, the research will lead us to the correlation between global trend and practice in cruise shipping and the safety and sustainability of the cruising. In the reality I cannot find his correlation. There was no mathematical correlation computed, nor quality analysis performed. No measures have been used. Please explain the relation to the tile in the text.

4. From my point of view, any state-of-the-art list of global trends, as well as practices should be listed as summary or at the end of introduction.

Author Response

Good afternoon Mrs. or Mr.

Thank your for your review and feed back received on my study. Please find attached corrections and reply related to your review.

Best Regards

Josip Dorigatti

Round 2

Reviewer 1 Report

i am satisfied with the corrections that have been made by the author. I suggest that this article be taken to the next level.

Reviewer 2 Report

Thank you for clarifications and paper improvement.